# Prediction of Gastric Gastrointestinal Stromal Tumors before Operation: A Retrospective Analysis of Gastric Subepithelial Tumors

**DOI:** 10.3390/jpm12020297

**Published:** 2022-02-17

**Authors:** Yu-Ning Lin, Ming-Yan Chen, Chun-Yi Tsai, Wen-Chi Chou, Jun-Te Hsu, Chun-Nan Yeh, Ta-Sen Yeh, Keng-Hao Liu

**Affiliations:** 1Department of General Surgery, Chang Gung Memorial Hospital at Linkou, Taoyuan 33305, Taiwan; lillian292@hotmail.com (Y.-N.L.); hitsuzi@cgmh.org.tw (M.-Y.C.); m7202@cgmh.org.tw (C.-Y.T.); hsujt2813@cgmh.org.tw (J.-T.H.); ycn@cgmh.org.tw (C.-N.Y.); tsy471027@cgmh.org.tw (T.-S.Y.); 2Department of Oncology, Chang Gung Memorial Hospital at Linkou, Taoyuan 33305, Taiwan; f12986@cgmh.org.tw

**Keywords:** diagnosis, gastrointestinal stromal tumor, gastric subepithelial tumor, stomach

## Abstract

Gastrointestinal stromal tumors (GISTs), leiomyomas, and schwannomas are the most common gastric subepithelial tumors (GSETs) with similar endoscopic findings. Preoperative prediction of GSETs is difficult. This study analyzed and predicted GSET diagnosis through a retrospective review of 395 patients who underwent surgical resection of GISTs, leiomyomas, and schwannomas measuring 2–10 cm. GSETs were divided by size (group 2–5, >2 and ≤5 cm; group 5–10, >5 and ≤10 cm) for analysis. Demographics, clinical symptoms, and images were analyzed. A recursive partitioning analysis (RPA) was used to identify optimal classifications for specific GSET diagnoses. GIST patients were relatively older than other patients. Both groups had higher proportions of UGI bleeding, lower hemoglobin (Hb) levels, and a higher ratio of necrosis on their computed tomography (CT) scans. The RPA tree showed that (a) age ≤ 55, Hb ≥ 10.7, and CT necrosis; (b) age ≤ 55 and Hb < 10.7; (c) age >55 and Hb < 12.9; and (d) age >55 and CT hetero-/homogeneity can predict high GIST risk in group 2–5. Positive or negative CT necrosis, with age >55, can predict high GIST risk in group 5–10. GIST patients were older and presented with low Hb levels and tumor necrosis. In RPA, the accuracy reached 85% and 89% in groups 2–5 and 5–10, respectively.

## 1. Introduction

Gastric subepithelial tumors (GSETs) are common findings in patients undergoing esophagogastroduodenoscopy (EGD). Gastric subepithelial lesions (SELs) can be divided into benign lesions, potentially malignant lesions, and malignant lesions. Benign lesions include leiomyoma, schwannoma, lipoma, and ectopic pancreas. Malignant or potentially malignant GSETs include gastrointestinal stromal tumors (GISTs), lymphomas, carcinoid tumors, and glomus tumors [1]. Different tumor characteristics, prevalence, and treatment planning depend on the pathologic diagnosis.

GISTs are the most common GSETs in the stomach, followed by leiomyomas and schwannomas [2]. These three GSETs are usually found in the muscularis propria layer by endoscopic ultrasound (EUS) [3]. However, approximately 10–30% of GISTs have malignant clinical courses [4,5]. The prognosis of GISTs is associated with the tumor size and mitotic index [3]. Despite advanced diagnostic equipment and studies, GSETs are still difficult to diagnose with non-invasive methods, such as EGD, EUS, and computed tomography (CT), preoperatively [6]. EUS biopsy is recognized as a better diagnostic tool. Some studies have reported that the diagnostic rate of endoscopic ultrasound-guided fine-needle aspiration (EUS-FNA) ranges from 62% to 80.6% [1,3,7]. However, EUS biopsy is not widely applied in GSETs and has limitations, such as insufficient specimens and small lesions. Because of the difficulty in predicting diagnosis and malignant potential, recent guidelines recommend surgical resection for gastric GISTs larger than 2 cm [6,8,9].

This study aimed to analyze and predict diagnosis based on preoperative clinical characteristics, laboratory results, and imaging studies among GISTs, leiomyomas, and schwannomas to refine the treatment strategy.

## 2. Materials and Methods

### 2.1. Study Design

This retrospective study was approved by the Institutional Review Board of Chang Gung Memorial Hospital (IRB No. 202100954B0). From January 2003 to December 2016, patients who were diagnosed with gastric GIST, leiomyoma, and schwannoma in Chang Gung Memorial Hospital at Linkou, a tertiary medical center in Taiwan, were enrolled and retrospectively reviewed. According to the modified National Institutes of Health’s risk of recurrence [9], tumors were divided into four groups: ≤2 cm, >2 and ≤5 cm (group 2–5), >5 and ≤10 cm (group 5–10), and >10 cm (group 10), for recording. Moreover, the tumors were divided into two groups, namely, GIST and leiomyoma/schwannoma, for analysis. Patients with GSETs who underwent biopsy only without further endoscopic or surgical resection, distant metastasis, and pathological tumor size <2 cm were excluded from the analysis. A total of 519 patients were diagnosed with GIST, leiomyoma, and schwannoma. Patient demographics and clinical symptoms, including age, sex, epigastric pain, upper gastrointestinal bleeding, body weight loss, abdominal fullness, dysphagia, vomiting, abdominal mass, obstruction symptoms, and incidental findings without symptoms, were recorded. Laboratory examinations (including hemoglobin (Hb) level and platelet count), liver function (including aspartate aminotransferase, alanine aminotransferase, and total and direct bilirubin levels), renal function, and tumor markers (including carcinoembryonic antigen and carbohydrate antigen 19-9) were recorded. The EGD and EUS findings included tumor size, location (esophagogastric junction (ECJ)/cardia/high body, middle body, and lower body/antrum/pylorus), mucosal surface, shape, with/without infiltrated border, and necrosis. The CT findings included ulceration, enhancement (heterogeneity/homogeneity), calcification, necrosis, adjacent organ involvement, and lymph node (LN) enlargement.

### 2.2. Statistical Analysis

All statistical analyses were performed using IBM SPSS for Windows, version 22 (IBM Corp., Armonk, NY, USA). Independent t-tests were conducted for continuous variables, and 95% confidence intervals were calculated. For categorical variables, Pearson’s chi-square test was used. A two-sided *p*-value of <0.05 was considered significant. Recursive partitioning, a statistical method for multivariable analysis, was used to create a decision tree that strives to correctly classify members of the population by splitting them into subpopulations based on several dichotomous independent variables. Significant variables identified in the univariate analysis were candidates for defining in the recursive partitioning analysis (RPA) tree to identify optimal patient classifications for specific GSET diagnoses.

## 3. Results

A total of 519 patients were diagnosed with gastric GSETs including 425 GISTs, 56 leiomyomas, and 38 schwannomas during the study period. The distribution of the tumor size is listed in Figure 1. Since only one schwannoma and no leiomyoma were diagnosed in group 10, this group was excluded from the analysis. A total of 395 patients were included in the analysis. As mentioned in the Materials and Methods, patients were divided into two groups according to the lesion sizes (group 2–5 and group 5–10) and compared between GISTs and non-GISTs (leiomyoma and schwannoma) for analysis.

Table 1 shows the proportion and analysis of basic data, clinical symptoms, and laboratory exams of GIST, leiomyoma, and schwannoma.

Regarding basic data, a significantly higher age was found in patients with GISTs in group 2–5 (63 years, *p* < 0.0001) and group 5–10 (61 years, *p* = 0.025), and significance was also shown at the age cut-off point of 55 years.

For clinical symptoms, the incidence of UGI bleeding was higher in group 2–5 (33.6%), which was significantly higher (*p* = 0.014) than the other two non-GIST GSETs (leiomyoma and schwannoma). In group 5–10, the incidence of UGI bleeding was also significantly higher in GISTs (35%, *p* = 0.035).

In the laboratory analysis, the Hb level was significantly lower (*p* < 0.0001) in group 2–5 patients with GIST (12.5 g/dL) than in patients with leiomyoma/schwannoma (13.8 g/dL). In group 5–10, the same result was noted. A significantly lower Hb level was found in patients with GIST (12 g/dL) compared with those with leiomyoma/schwannoma (13.6 g/dL) (*p* = 0.014).

The EGD, EUS, and CT findings are presented in Table 2. In group 2–5, 85 cases were located in the ECJ/cardia/high body area, 14 in the middle body, and 26 in the lower body/antrum/pylorus, and 98 cases did not belong to any locations mentioned. In group 5–10, 35 cases were located in the ECJ/cardia/high body area, 12 in the middle body, and 10 in the lower body/antrum/pylorus, and 46 cases did not belong to these locations. Although a slightly higher proportion of GISTs were located in the upper stomach, no significant difference was found in EGD findings between GIST and leiomyoma/schwannoma.

Regarding EUS findings, significant differences were found between GIST and leiomyoma/schwannoma in the location of the layer and echogenicity in groups 2–5 and 5–10.

The CT findings are summarized in Table 2. No significant differences were observed in calcification, ulceration, adjacent organ involvement, or lymph node enlargement. In groups 2–5 and 5–10, a significant difference was found in the enhancement pattern, with *p* = 0.003 and 0.029, respectively. Tumor necrosis was more often observed in GISTs. Tumor necrosis was noted in 21.5% in group 2–5, and 59.2% in group 5–10 GIST patients. Significance between GISTs and leiomyoma/schwannoma was noted in both groups (*p* = 0.014 and *p* < 0.001).

Based on the result of the significant prognostic variables, RPA was performed to reclassify the decision tree in group 2–5 (Figure 2) and group 5–10 (Figure 3).

To integrate and simplify the RPA, Table 3 organizes the patient characteristics of nodes in each group, patient numbers, and prediction accuracy.

In the group 2–5 RPA tree, a total of 21 terminal nodes were produced in 10 splits (Figure 2). Among the variables included in the analysis, upper GI bleeding was omitted because we used the Hb level for the analysis. The terminal nodes were categorized into groups I to IV based on their predictive ability for GISTs. Group I had the lowest probability (incidence 39%; 95% CI, 37.5–40.0%), while group IV had the highest probability (incidence 95.5%; 95% CI, 88.9–96.1%; *p* < 0.001), for being diagnosed with GISTs. Patients with age ≤ 55, Hb ≥ 10.7, and necrosis findings on CT (node 13); age ≥ 55 and Hb < 10.7 (node 14); age >55 without CT-specific findings but with Hb < 12.9 (node 20); and age >55 and CT heterogeneity and homogeneity (node 21) can expect the highest predictive ability of GISTs (group IV).

In the group 5–10 RPA tree, a total of three terminal nodes were produced in two splits (Figure 3). Patients with negative CT necrosis plus age ≤ 55 had the lowest probability of being diagnosed with GISTs (group I, incidence 20.0%). Patients with negative CT necrosis but with age > 55 had an intermediate probability of being diagnosed with GISTs (group II, incidence 81.8%, *p* = 0.003 when compared with group I). Patients with positive CT necrosis had the highest probability of being diagnosed with GISTs (group III, incidence 91.8%, *p* < 0.001 when compared with group I).

Significant differences were observed between the reference group (group I) and other groups according to tumor size 2–5 cm and 5–10 cm RPA, with ORs ranging from 2.34 to 47.43 (Table 4).

Table 4 summarizes the risk classification according to the RPA results. Using the present results, predicted probabilities can be estimated based on these pre-treatment examinations.

## 4. Discussion

GSETs are commonly observed on EGD. GISTs, leiomyomas, and schwannomas are the three most common types of GSETs. Because of the uncertain behavior of GISTs, the National Comprehensive Cancer Network guidelines suggest that GISTs with a size ≥ 2 or < 2 cm with symptoms should be removed [9]. The European Society for Medical Oncology group proposed that histologically diagnosed small GISTs should be removed [10]. Preoperative prediction of GISTs is important in GSETs. Despite advanced imaging studies and endoscopic equipment, making an accurate diagnosis using preoperative non-invasive modalities remains difficult.

A total of 519 GSETs were diagnosed at our institute. Regarding the tumor size, GSETs had the highest proportion in group 2–5 (*n* = 273, 53.2%), followed by group 5–10 (*n* = 122, 23.8%). GIST was the most common diagnosis (82.8%) of all GSETs, followed by leiomyoma (9.9%) and schwannoma (7.3%), and the diagnostic proportion of the three GSETs was similar to that reported in a previous study [11]. We excluded GSETs ≥ 10 cm for the analysis because most of the tumors in this group were GISTs, except for one schwannoma.

Some clinical difficulties for endoscopic biopsies, such as tumors beyond the overlying normal mucosa, which might not be reached, insufficient tissue sample volume, and the possibility of procedure-related complications, such as bleeding and stomach perforation, were observed. Preoperative direct tissue sampling methods, such as conventional endoscopic forceps biopsy, EUS-FNA biopsy, and jumbo biopsy, might not provide sufficient information for diagnosis [2,3,12,13]. Bang et al. reported a meta-analysis comparing ProCore and standard FNA needles for EUS tissue acquisition. The accuracy of an EUS-FNA or EUS-Tru-cut biopsy ranges from 34% to 79% [13,14,15]. Even with a newly developed ProCore needle, the diagnostic rate could only be increased to 81.8% [16]. We did not include the EGD and EUS biopsy data for analysis because biopsy was not routinely performed in all patients with GSETs in our institute, and false negatives were commonly seen if EUS-FNA biopsy or jumbo biopsy was not performed.

Our results show that patients with GISTs were usually older (group 2–5, 63 years; group 5–10, 61 years), which is in agreement with previous studies [2,17,18]. The most common presentation of GISTs is bleeding-related mucosal erosion. As previously reported, GISTs commonly present with bleeding-related mucosal erosion [19]. A certain proportion of our patients experienced symptoms of gastrointestinal bleeding and ulceration (group 2–5, 33.6%; group 5–10 cm, 35%) in the EGD. In addition to the clinical symptoms and EGD findings, we also found that GISTs had a significantly higher proportion of anemia in groups 2–5 and 5–10 compared with leiomyoma and schwannoma. This suggests that aggressive management of GISTs is mandatory to prevent further bleeding-related complications.

EUS provides high-resolution tomographic imaging using high-frequency ultrasound for differential diagnosis of SELs. A typical EUS imaging feature of GIST is a hypoechoic solid mass with a size >2 cm, irregular borders, heterogeneous echo patterns, anechoic spaces, echogenic foci, and growth during follow-up [20,21]. However, malignant lymphoma, metastatic cancer, neuroendocrine tumor, and SEL-like cancer and benign conditions (such as leiomyoma, neurinoma, or aberrant pancreas) also presented with a hypoechoic solid mass [3]. An accurate diagnosis of SELs using only EUS is difficult, with accuracy ranging from 45.5% to 48.0% according to a recent study [22]. Tissue sampling for immunohistochemical analysis using EUS-FNA or biopsy should be considered for further differential diagnosis of SEL. Although EUS tissue acquisition (EUS-TA) is not routinely performed for GSETs in our hospital, with increasing experience, it may be safe and may provide helpful information for further treatment planning in clinical practice.

CT features for benign GSETs are generally nonspecific, but they have other advantages in evaluating the characteristics, local extension, invasion to adjacent organs, or possible metastasis of GSETs [1,23]. In studies comparing CT findings, true gastric leiomyoma has homogeneous enhancement, poor to moderate enhancement, and low attenuation or isoattenuation compared with muscle attenuation [1,24]. Gastric schwannomas are relatively uncommon among GSETs. Ji et al. reported that gastric schwannomas usually present with ovoid, well-defined, exophytic, or mixed growth on CT [25]. However, GISTs frequently present with irregular margins, heterogeneous enhancement, and necrosis [18]. In our study, similar CT characteristics were observed. Most leiomyomas presented with a higher probability of homogeneity enhancement, and tumor necrosis was significantly higher (*p* < 0.001) in group 5–10 GISTs (59.2%).

To derive a diagnostic model of GSETs with preoperative symptoms and clinical data from laboratory and imaging modalities, we included the significant variables in the univariate analysis into the RPA. Several clinical studies employ RPA to define risk groups [26,27,28]. RPA is a tool for the stratification of risk or prognostic factors and identification of a homogenous group of patients [29]. In this study, age, hemogram, and CT findings such as necrosis and hetero-/homogeneity were significant variables for predicting tumor type among patients in group 2–5. In 2–5 cm GSETs, patients (1) with age ≤ 55, Hb ≥ 10.7, and necrosis findings on CT, (2) age ≤ 55 and Hb < 10.7, (3) age >55, without CT-specific findings but with Hb < 12.9, and (4) age > 55 and CT heterogeneity and homogeneity can expect the highest prediction ability of GISTs. On the contrary, younger patients (age ≤ 55) with normal Hb levels have a lower chance of being diagnosed with GISTs. The results suggest that gastrointestinal tract bleeding might serve as an important clinical clue for the diagnosis of GISTs compared to the other two tumor types in cases of tumor sizes ≥ 2 and < 5 cm, whereas gastrointestinal tract bleeding is a common phenomenon for tumor size ≥ 5 cm regardless of the tumor type. The presence of tumor necrosis detected by imaging and age > 55 were significant predictors of GISTs in group 5–10 according to the RPA. Tumor size is a well-known prognosticator for patients with GISTs, indicating a higher malignant potential, aggressive tumor behavior, and poor tumor prognosis. By contrast, tumor size was not associated with tumor behavior in patients with schwannomas or leiomyomas. Tumor necrosis is a surrogate for rapid tumor growth and aggressive tumor behavior.

Based on our study, for patients who have a higher probability of GISTs, aggressive surgical treatment with an adequate surgical margin is mandatory.

This study had several limitations. First, this was a single-center, retrospective, observational study. Second, except for GISTs, the sample sizes of the other two tumors were small. Third, we included the three most common diagnoses of GSETs and excluded other relatively rare diseases such as lymphoma, ectopic pancreas, and duplication cyst. Fourth, our study only included GSETs with a size ≥ 2 cm, which suggests surgical treatment from the surgeon’s perspective. Gastric tumors < 2 cm might be under periodic surveillance with EGD or EUS. Biopsy or excision was not routinely performed in this group of patients; therefore, GSETs < 2 cm were excluded from our study. Fifth, we also excluded GSETs ≥ 2 cm without definite pathology reports acquired from surgery or biopsy. Finally, selection bias might exist, and this might be the weak point of the study. Thus, expansion of the database and interdisciplinary analysis may be needed in our future research.

## 5. Conclusions

We retrospectively analyzed the preoperative clinical characteristics and imaging findings to predict the diagnosis of GSETs. Patients who were diagnosed with GISTs were usually older, presented with low Hb levels, and showed tumor necrosis on CT. In the RPA result of our study, the clinical findings and preoperative image findings could help predict diagnosis, which could assist surgeons in treatment planning before operation and other management.

## Figures and Tables

**Figure 1 jpm-12-00297-f001:**
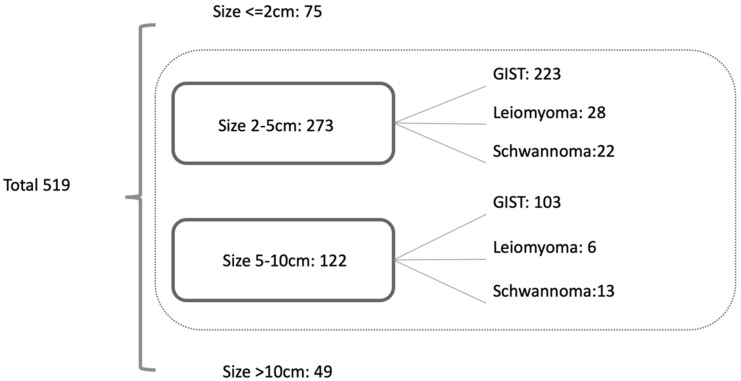
Distribution of GSETs.

**Figure 2 jpm-12-00297-f002:**
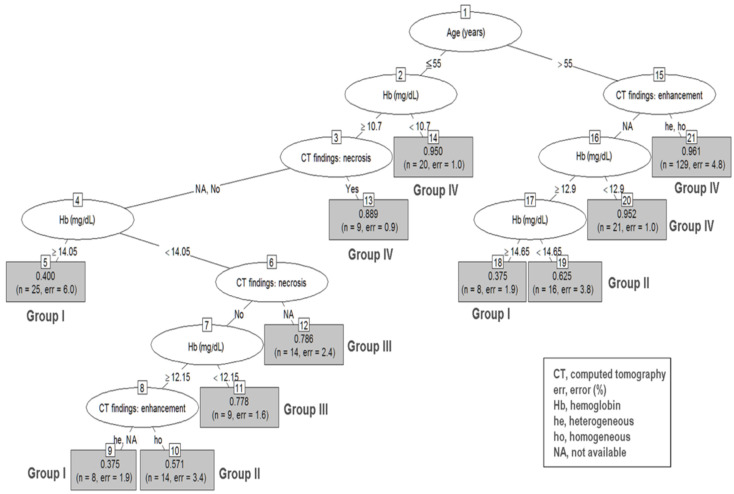
Recursive partitioning analysis of group 2–5 gastric subepithelial tumors. Decision tree constructed by recursive partitioning analysis in group 2–5 with 0.85 prediction accuracy. Terminal nodes are categorized into groups I to IV based on their prediction ability for gastrointestinal stromal tumors. Group I: nodes 5, 9, and 18 (0.375–0.400); group II: nodes 10 and 19 (0.571–0.625); group III: nodes 11 and 12 (0.778–0.786); group IV: nodes 13, 14, 20, and 21 (0.889–0.961).

**Figure 3 jpm-12-00297-f003:**
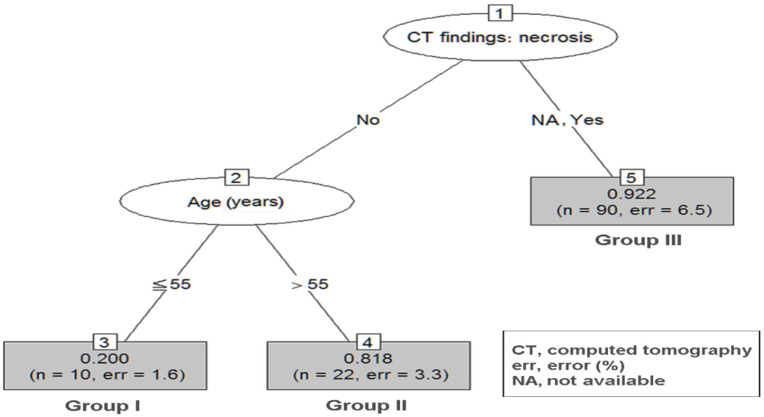
RPA of group 5–10 GSETs. Decision tree constructed by RPA in group 5–10. The prediction accuracy in RPA is 0.89. GSETs, gastric subepithelial tumors; RPA, recursive partitioning analysis.

**Table 1 jpm-12-00297-t001:** Basic data, symptoms, and laboratory results.

Characteristics	Group 2–5			Group 5–10		
	GIST (*n* = 223)	Leiomyoma/ Schwannoma (*n* = 50)	*p* Value	GIST (*n* = 103)	Leiomyoma/ Schwannoma (*n* = 19)	*p* Value
**Basic data**						
Age (years)	63 (19)	50 (21)	<0.0001	61 (18)	52 (20)	0.025
≦55	66 (29.6)	33 (66.0)	<0.0001	31 (30.1)	13 (68.4)	0.001
>55	157 (70.4)	17 (34.0)		72 (69.9)	6 (31.6)	
Sex			0.267			0.193
Male	104 (46.6)	19 (38.0)		60 (58.3)	8 (42.1)	
Female	119 (53.4)	31 (62.0)		43 (41.7)	11 (57.9)	
**Symptoms**						
Epigastric pain	67 (30.0)	15 (30.0)	1.000	26 (25.2)	8 (12.1)	0.132
UGI bleeding	75 (33.6)	8 (16.0)	0.014	36 (35.0)	2 (10.5)	0.035
Body weight loss	1 (0.4)	0	>0.999	3 (2.9)	0	>0.999
Fullness	28 (12.6)	7 (14.0)	0.783	18 (17.5)	2 (10.5)	0.736
Dysphagia	5 (2.2)	1 (2.0)	>0.999	1 (1.0)	0	>0.999
Vomiting	12 (5.4)	3 (6.0)	0.742	5 (4.9)	0	>0.999
Abdominal mass	3 (1.3)	0	>0.999	5 (4.9)	1 (5.3)	>0.999
Obstruction	1 (0.4)	0	>0.999	0	0	n/a
Incidental finding	43 (19.3)	14 (28.0)	0.170	21 (20.4)	5 (26.3)	0.551
**Laboratory data**						
Hemoglobin (g/dL)	12.5 (3.5)	13.8 (2.3)	<0.0001	12.0 (3.4)	13.6 (1.6)	0.014
Platelet (10^3^/µL)	229.5 (96)	244 (62)	0.356	234.5 (99)	252 (138)	0.415
AST (µL)	21.5 (10)	20.5 (7)	0.261	20 (10)	19 (15)	0.468
ALT (U/L)	18.0 (12.0)	19.5 (10.0)	0.665	19.0 (14.0)	17.5 (15.0)	0.817
Bilirubin total (mg/dL)	0.5 (0.3)	0.5 (0.2)	0.549	0.5 (0.4)	0.5 (0.3)	0.839
BUN (mg/dL)	15.0 (6.1)	14.5 (6.5)	0.989	13.8 (6.8)	14.7 (10.9)	0.448
Creatinine (mg/dL)	0.8 (0.3)	0.8 (0.3)	0.916	0.9 (0.4)	0.8 (0.6)	0.940

Data are presented as median (IQR) or number (%). UGI, upper gastrointestinal.

**Table 2 jpm-12-00297-t002:** EGD, EUS, and CT finding analysis.

Characteristics	Group 2–5			Group 5–10		
	GIST (*n* = 223)	Leiomyoma/ Schwannoma (*n* = 50)	*p* Value	GIST (*n* = 103)	Leiomyoma/ Schwannoma (*n* = 19)	*p* Value
**EGD findings**						
Tumor location			0.829			0.410
ECJ/cardia/high body	85 (38.1	18 (36.0)		35 (34.0)	4 (21.1)	
Middle body	14 (6.3)	3 (6.0)		12 (11.7)	3 (15.8)	
Lower body/antrum/pylorus	26 (11.7)	4 (8.0)		10 (9.7)	4 (21.1)	
NA	98 (43.9)	25 (50.0)		46 (44.7)	8 (42.1)	
Ulcer			0.553			0.287
No	56 (25.1)	14 (28.0)		18 (17.5)	6 (31.6)	
Yes	71 (31.8)	12 (24.0)		42 (40.8)	5 (26.3)	
Unknown/not done	96 (43.0)	24 (48.0)		43 (41.7)	8 (42.1)	
Infiltration			0.238			0.689
No	124 (55.6)	22 (44.0)		55 (53.4)	10 (52.6)	
Yes	2 (0.9)	0		2 (1.9)	1 (5.3)	
NA	97 (43.5)	28 (56.0)		46 (44.7)	8 (42.1)	
**EUS findings**						
Layer			0.283			0.965
Muscularis propria	120 (53.8)	33 (66.0)		32 (31.1)	6 (31.6)	
Submucosa	8 (3.6)	1 (2.0)		0	0	
Unknown/not done	95 (42.6)	16 (32.0)		71 (68.9)	13 (68.4)	
Echogenicity			0.162			0.781
Hyper	0	0		1 (1.0)	0	
Hypo	114 (51.1)	34 (68.0)		26 (25.2)	6 (31.6)	
Iso	2 (0.9)	0		0	0	
Mixed	2 (0.9)	0		0	0	
NA	105 (47.1)	16 (32.0)		76 (73.8)	13 (68.4)	
**CT findings**						
Enhancement			0.003			0.029
Heterogeneity	60 (26.9)	6 (12.0)		53 (51.5)	4 (21.1)	
Homogeneity	110 (49.3)	21 (42.0)		30 (29.1)	7 (36.8)	
NA	53 (23.8)	23 (46.0)		20 (19.4)	8 (42.1)	
Calcification			0.154			0.912
No	146 (65.5)	32 (64.0)		70 (68.0)	12 (63.2)	
Yes	26 (11.7)	2 (4.0)		15 (14.6)	3 (15.8)	
NA	51 (22.9)	16 (32.0)		18 (17.5)	4 (21.1)	
Ulcer			0.298			0.718
No	132 (59.2)	30 (60.0)		54 (52.4)	11 (57.9)	
Yes	40 (17.9)	5 (10.0)		31 (30.1)	4 (21.1)	
NA	51 (22.9)	15 (30.0)		18 (17.5)	4 (21.1)	
Necrosis			0.014			<0.001
No	121 (54.3)	32 (64.0)		20 (19.4)	12 (63.2)	
Yes	48 (21.5)	2 (4.0)		61 (59.2)	3 (15.8)	
NA	54 (24.2)	16 (32.0)		22 (21.4)	4 (21.1)	
Adjacent organ involvement			0.300			0.096
No	171 (76.7)	34 (68.0)		64 (62.1)	15 (78.9)	
Yes	2 (0.9)	0		21 (20.4)	0	
NA	50 (22.4)	16 (32.0)		18 (17.5)	4 (21.1)	
LN enlargement			0.120			0.310
No	173 (77.6)	32 964.0)		74 (71.8)	12 (63.2)	
Yes	4 (1.8)	2 (4.0)		6 (5.8)	3 (15.8)	
NA	46 (20.6)	16 (32.0)		23 (22.3)	4 (21.1)	

Abbreviation: EGD, esophagogastroduodenoscopy; EUS, endoscopic ultrasound; CT, computed tomography; NA, not available. Data are presented as number (%).

**Table 3 jpm-12-00297-t003:** Risk classification according to the results of the recursive partitioning analysis.

	Nodes	Risk Groups	Number of Patients	Prediction Accuracy
**Size 2–5 cm**				
Group I	5	age ≦ 55, CT necrosis: no/NA, Hb ≧ 14.05	25	0.400
	9	age ≦ 55, no CT necrosis, CT heterogeneity/NA,12.15 ≦ Hb < 14.05	8	0.375
	18	age > 55, no CT enhancement, Hb ≧ 14.65	8	0.375
Group II	10	age ≦ 55, no CT necrosis, CT homogeneity, 12.15 ≦ Hb < 14.05	14	0.571
	19	age > 55, no CT enhancement, 12.9 ≦ Hb < 14.65	16	0.625
Group III	11	age ≦ 55, no CT necrosis, 10.7 ≦ Hb < 12.15	9	0.778
	12	age ≦ 55, CT necrosis: NA, 10.7 ≦ Hb < 14.05	14	0.786
Group IV	13	age ≦ 55, Hb ≧ 10.7, CT necrosis	9	0.889
	14	age ≦ 55, Hb < 10.7	20	0.950
	20	age > 55, CT enhancement: NA, Hb < 12.9	21	0.952
	21	age > 55, CT heterogeneity/homogeneity	129	0.961
**Size 5–10 cm**				
Group I	3	age ≦ 55, no CT necrosis	10	0.200
Group II	4	age > 55, no CT necrosis	22	0.818
Group III	5	CT necrosis: yes/NA	90	0.922

**Table 4 jpm-12-00297-t004:** Risk classification according to the results of the recursive partitioning analysis.

Risk Classification	GIST		Odds Ratio	95% CI	*p* Value
	Yes	No			
Size 2–5 cm					
Group I	16 (39.0)	25 (61.0)	1		
Group II	18 (60.0)	12 (40.0)	2.34	0.90–6.14	0.083
Group III	18 (78.3)	5 (21.7)	5.63	1.74–18.17	0.004
Group IV	171 (95.5)	8 (4.5)	33.40	12.96–86.08	<0.0001
Size 5–10 cm					
Group I	2 (20.0)	8 (80.0)	1		
Group II	18 (81.8)	12 (18.2)	18.00	2.72–119.23	0.003
Group III	83 (92.2)	8 (7.8)	47.43	8.40–267.77	<0.0001

## Data Availability

Not applicable.

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
