# Peer review of "Prediction of Gastric Gastrointestinal Stromal Tumors before Operation: A Retrospective Analysis of Gastric Subepithelial Tumors"

_jpm, 2022, doi:10.3390/jpm12020297_

Round 1

Reviewer 1 Report

This is a large retrospective study using recursive partitioning analysis to predict a diagnosis of GIST in patients with a gastric subepithelial lesion.

In the abstract, the group division is not clear. Please, check this sentence: “GSETs were divided by size (>2, ≤5cm, group2-5; and >5, ≤10 cm, group5-10) for analysis.

A validation (internal or external) of the model should be provided.

The decision tree of the group "2-5" seems very complex. Could it be simplified? As presented in this paper it will be very difficult to be used in clinical practice.

A deeper comment on the role of EUS appearance and EUS-biopsy should be included. Doing so, cite PMID: 31597179.

Author Response

Response to Reviewer 1 Comments

Point1. In the abstract, the group division is not clear. Please, check this sentence: “GSETs were divided by size (>2, ≤5cm, group2-5; and >5, ≤10 cm, group5-10) for analysis.

Response1: We changed the sentence to “GSETs were divided by sizes (group2-5: >2, and ≤5cm; and group5-10:>5, and ≤10 cm) for analysis." And we marked the sentence with yellow color in the manuscript.

Point 2. A validation (internal or external) of the model should be provided.

Response2:
Thank you very much for this great suggestion. Because of the IRB of this study did not apply the permission for the data acquisition and we did not have the patient data after 2016, validation of the model would be difficult and cannot be performed. And because of the time limitation (uploading revised file in 10 days), we did not have enough time to acquire the data for validation. Validation could be used and considered in our future study.

Point 3. The decision tree of the group "2-5" seems very complex. Could it be simplified? As presented in this paper it will be very difficult to be used in clinical practice.

Response 3:
Thank you for the suggestion. The decision tree seems complex, but without this RPA analysis, the risks groups, and odds ratio might lose their integrity. To integrate and simplify the decision tree, Table 3 organized the patient characteristic of nodes in each group, patient numbers, and prediction accuracy to make it more clear. And we add this paragraph and marked with yellow in the manuscript.

Point 4. A deeper comment on the role of EUS appearance and EUS-biopsy should be included. Doing so, cite PMID: 31597179.

Response 4:

Thank you for the suggestion. We add this paragraph : "EUS provides high-resolution tomographic imaging using high-frequency ultrasound for differential diagnosis of SELs. Typical EUS imaging feature of GIST is a hypoechoic solid mass with size >2 cm, irregular borders, heterogeneous echo patterns, anechoic spaces, echogenic foci, and growth during follow-up. However, malignant lymphoma, metastatic cancer, neuroendocrine tumor, and SEL-like cancer and benign conditions (such as leiomyoma, neurinoma, or aberrant pancreas) also presented with a hypoechoic solid mass. The accurate diagnosis of SELs with EUS only is difficult, with accuracy ranging from 45.5% to 48.0% according to a recent study. Tissue sampling for immunohistochemical analysis using EUS-FNA or biopsy should be considered for further differential diagnosis of SEL. Although EUS tissue acquisition (EUS-TA) is not routinely performed for GSETs in our hospital, with increasing experience, it may be safe and provide helpful information for further treatment planning in clinical practice.”  in discussion and marked with yellow background.

Point 5.  English language and style are fine/minor spell check required  

Answer:
Minor spell and grammar error were revised. And Editage had helped with the English fluency again. 

Reviewer 2 Report

The paper analyzes a critical number of patients resected for gastric subepithelial tumors in a significant single-center experience. The paper is well-written, the analyses are correctly performed, and the provided results support the conclusions. The paper would be of real interest to the readers of the journal.

Few minor comments:

Please explain involvement in Table 2.

The paper needs a few revisions for English fluency by a native speaker or a professional editing service.

Author Response

Response to Reviewer 2 Comments  

Point 1. Please explain involvement in Table 2.

Response 1:

Involvement means adjacent organ involvement on CT. We’ve revised and marked with yellow background in Table 2.

Point 2.The paper needs a few revisions for English fluency by a native speaker or a professional editing service.

Response 2: Thank you for the suggestion. The manuscript is reviewed and revised by Editage again. The certificate is also attached in the file below.

Round 2

Reviewer 1 Report

I have no further comments